# LINSCAN - A Linearity Based Clustering Algorithm

## Abstract

DBSCAN and OPTICS are powerful algorithms for identifying clusters of points in domains where few assumptions can be made about the structure of the data. In this paper, we leverage these strengths and introduce a new algorithm, LINSCAN, designed to seek lineated clusters that are difficult to find and isolate with existing methods. In particular, by embedding points as normal distributions approximating their local neighborhoods and leveraging a distance function derived from the Kullback Leibler Divergence, LINSCAN can detect and distinguish lineated clusters that are spatially close but have orthogonal covariances. We demonstrate how LINSCAN can be applied to seismic data to identify active faults, including intersecting faults, and determine their orientation. Finally, we discuss the properties a generalization of DBSCAN and OPTICS must have in order to retain the stability benefits of these algorithms.

## 1 Introduction

Many existing clustering algorithms require some prior knowledge of the dataset and are limited in the possible shapes they can identify. For example, both K-Means Clustering and Gaussian Mixture Model (GMM) Expectation Maximization require a prior estimate of the number of clusters existing in the dataset and struggle to distinguish clusters that are not linearly separable.

In contrast, DBSCAN and OPTICS iteratively generate clusters by leveraging a heuristic for the local behavior of clustered points. In particular, the designers equated clusters to connected regions of high density (Ester et al., 1996). Thus, by identifying points whose local neighborhoods are highly dense, even with little prior knowledge about the local geometry of the data, one can iteratively grow clusters from those points. The number of clusters then comes naturally from the geometry of the data itself, rather than being a parameter.

In this paper, we seek to leverage this characterization of clusters using a clustering metric other than Euclidean distance. In particular, we propose an algorithm that can distinguish between multiple quasi-linear clusters that may be closely spaced but have nearly orthogonal covariances. This is motivated in particular by the need to identify and map seismically active faults given a catalog of precisely located earthquakes, an important problem in geophysics (Fialko, 2021; Zou et al., 2023; Shelly et al., 2023). In addition, the potential of the algorithm is not limited to geophysics, but it may also help identify the linear spatial patterns of other natural features such as soil and airborne pollution, and man-made directional patterns including roads and hiking trials (Barden, 1963; Isaaks and Srivastava, 1989; Mai et al., 2018).

### 1.1 Motivating Problem

We wish to isolate quasi-linear clusters (QLCs) in point clouds and distinguish clusters that are geometrically close, or possibly overlap, but have different orientations. Quasi linear clusters are a cluster of points where: 1) each point is within $\epsilon$ of some other point in the cluster, 2) the total

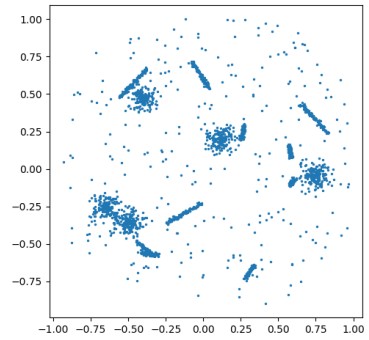 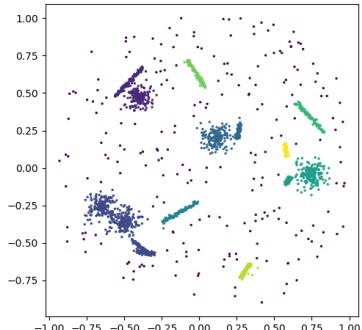

Figure 1: (Left) Test data, (Right) DBSCAN Results

cluster has a nearly singular covariance matrix. This problem arises, for example, in geophysics, when one attempts to identify active seismogenic faults based on epicentral locations of microearthquakes (Cochran et al., 2020; Fialko, 2021; Shelly et al., 2023). Although faults are three-dimensional quasi-planar surfaces, in appropriate projections they appear as linear features, so that the associated locations of micro-earthquakes can be recognized as quasi-linear features after accounting for noise.

To highlight the deficiency of existing algorithms for this task, consider a synthetic data set shown in Figure 1. The data set includes QLCs, some of which intersect each other (e.g., see around coordinate (-.4,-.6)), as well as irregularly shaped clusters and "background noise." Figure 1 shows the results obtained by applying DBSCAN (described in Section 2) to the data. Note that the output includes both linear and irregular clusters, with some QLCs conjoined with irregular clusters. Furthermore, many QLCs that are geometrically close (e.g., intersecting or overlapping) are considered to be part of the same cluster.

## 1.2 Contributions

    a. We design an algorithm that can be used to identify quasi-linear clusters in a point cloud without losing the stability guarantees of well-established clustering algorithms like DBSCAN and OPTICS.

    b. We compare our framework to ADCN (Mai et al., 2016), a previous attempt at applying DBSCAN to a similar task, and discuss how the design of ADCN leads to the shape and number of clusters being sensitive to changes in the order of the points. This is in contrast to LINSCAN, which is invariant to the ordering of the points for clustering.

    c. We prove that while our distance measure is not a metric, it satisfies positivity and symmetry on the space of Gaussian distributions (see Lemma 3.1), and a slightly relaxed form of the triangle inequality (see Theorem 3.2). These results combine to mean that clusters in this metric are stable under the order of the points and are spatially dense.

## 1.3 Notation

Here we summarize the notation that will be used throughout the rest of this paper:

    a. For $\epsilon > 0$, we let $B_\epsilon(x)$ be the open ball of radius $\epsilon$ centered at $x$ (in the standard Euclidean norm).

    b. For finite $E \subseteq \mathbb{R}^d$,

   (a) $\mu_E \in \mathbb{R}^d$ is the sample mean of $E$.

   (b) $\Sigma_E \in \mathbb{R}^{d \times d}$ is the sample covariance matrix of $E$.

   Given $\mu \in \mathbb{R}^d$ and $\Sigma \in \mathbb{R}^{d \times d}$ with $\Sigma$ symmetric positive definite, $\mathcal{N}(\mu, \Sigma)$ is the multivariate Gaussian distribution with mean $\mu$ and covariance $\Sigma$.

c. We let $\|A\|_F := \mathrm{tr}(A^T A)$ denote the Frobenius norm.

d. For positive definite $A$, $\|x\|_A := \sqrt{x^T A x}$ is the elliptic norm defined by $A$.

e. For a general matrix $A$, let $|A|$ denote the determinant of $A$.

f. We let $\mathbb{P}(\mathcal{X})$ be the set of probability distributions on $\mathcal{X}$ and let $\mathcal{P}(\mathcal{X})$ be the power set of $\mathcal{X}$.

g. A QLC is a subset of points $S \subset \mathcal{X}$ satisfying:

   (a) $\forall x \in S, \exists y \in S \setminus \{x\}$ such that $\|x - y\| < \epsilon$ for some small $\epsilon$,

   (b) the covariance $\Sigma_S$ satisfies $\mathrm{cond}(\Sigma_S) > \tau$ for some large $\tau$.

We begin by summarizing the most popular clustering algorithms, namely DBSCAN and OPTICS, to provide context for readers lacking a background in clustering theory. Those familiar with DBSCAN and OPTICS can skip directly to section 3.

## 2  Background: DBSCAN and OPTICS

### 2.1  DBSCAN

The main principle behind DBSCAN is that clusters are equivalent to connected regions of high density. Thus, the most natural way to identify clusters is to search for points whose local neighborhoods contain a high density of points from the dataset and inductively grow clusters from those points.

In what follows, assume $X = \{x_1, ..., x_m\} \subseteq \mathbb{R}^d$ is a point cloud and let $\epsilon > 0$ and minPts $\in \mathbb{N}$ be two parameters. We say $x \in X$ is a **core point** if $\#(B_\epsilon(x) \cap X) > $ minPts, where $B_\epsilon(x)$ is the ball of radius $\epsilon$ about $x$.

Then, for two points $p$ and $q$, we say $q$ is **core reachable** from $p$ if there exist core points $p_1, ..., p_n$ such that $p_{k+1} \in B_\epsilon(p_k)$ for all $k \in \{0, ..., n-1\}$, $p \in B_\epsilon(p_0)$, and $q \in B_\epsilon(p_n)$.

As a result, core reachability is an equivalence relation. DBSCAN then defines clusters to simply be equivalence classes under this relation, with clusters containing fewer than minPts points being labeled as noise. Algorithm 1 in the supplemental documents provides a pseudocode description of how this is done.

DBSCAN satisfies a few important properties. First, because core reachability is independent of the order of the points, DBSCAN is invariant under permutations of the point cloud. Furthermore, we do not need to specify the number of clusters beforehand, and all of the operations are highly efficient so long as one can efficiently calculate $B_\epsilon(x) \cap X$.

### 2.2  OPTICS

OPTICS acts as a generalization of DBSCAN, improving its robustness on datasets with regions of various densities and partially abstracting away the $\epsilon$ parameter (Ankerst et al., 1999). The most popular and effective implementation of OPTICS takes in three parameters: $\epsilon$, minPts, and $\xi$, although $\epsilon$ is optional and only serves to shorten the run-time of the algorithm.

For $p \in X$, let $R_\delta(p) := X \cap B_\delta(p)$ for $\delta > 0$. We let the **core distance** $d_{\mathrm{core}}(p)$ be the minimum $\delta$ such that $R_\delta(p)$ contains minPts points. Alternatively, it is the minimum $\delta$ such that $p$ would be considered a core point if DBSCAN were to be performed using $\delta$ as $\epsilon$.

For $p, o \in X$, we define the **reachability distance** from $o$ to $p$ as $d_{\mathrm{reach}}(p|o) = \max \{ d_{\mathrm{core}}(o), \|p - o\| \}$.

The reachability distance describes the minimum $\epsilon$ such that $o$ is considered a core point and $p$ is contained in an $\epsilon$-neighborhood of $o$. Note that this can be infinite if $d_{\mathrm{core}}(o) = \infty$. OPTICS proceeds to develop a priority queue using a process described in the supplemental document in Algorithms 2 and 3.

While OPTICS is slightly slower than DBSCAN, it abstracts away one of the parameters, replacing it with one less tied to the geometry of $X$. Furthermore, it is far more robust to datasets with regions of varying density.

## 2.3 Related Work

The choice to use Euclidean distance with DBSCAN/OPTICS is arbitrary. The stability of the algorithm only depends on the fact that the distance function is symmetric and non-negative. Importantly, the function does not need to satisfy the triangle inequality (e.g., Khamsi and Kirk, 2011, p. 8), which allows us to work with non-metrics.

**Anisotropic DBSCAN:** The idea of extending DBSCAN/OPTICS to domains where we seek linearity is not entirely new. Previously, an algorithm called ADCN was developed to solve this problem by redefining the search neighborhoods from circles to ellipses whose eccentricity reflects the local covariance of the point (Mai et al., 2016). In practice, ADCN performs as well as DBSCAN in many tasks and performs better in cases where clusters are locally linear in otherwise highly noisy datasets.

However, ADCN is not well-suited for our task in particular because it does not provide the desired separation of adjacent or intersecting QLCs. On the contrary, it can produce artifacts around the intersection areas, say for a T-shaped intersection as in Figure 3. Furthermore, the point selection process in ADCN is non-symmetric, meaning that in certain cases the clustering behavior may be unstable to permutations of the points. Figures 3d and 3e show two runs of ADCN on the same dataset with the same parameters but with the dataset in a different order. Note how sensitive the behavior of the algorithm is to the order of the points. Our proposed algorithm performs more stably, as demonstrated below.

**Anisotropic Kernels and Spectral Clustering:** There are a large number of kernel method algorithms that use anisotropic kernels and local Mahalanobis distances to define similarity, see for example Wang et al. (2007); Talmon and Coifman (2013); Arias-Castro et al. (2017); Lahav et al. (2019); Cheng et al. (2020); Peterfreund et al. (2020). In practice, these can capture a similar notion of local similarity to our proposed approach and have been used for spectral clustering. For example, Arias-Castro et al. (2017) considers a similar problem to ours in clustering data that arises from intersecting manifolds.

However, regardless of the kernel similarity, spectral clustering and k-means (or another clustering algorithm) in the latent space fail in our noisy setting, where most of the points do not belong to any cluster. This is because k-means and spectral clustering algorithms perform poorly for data sets that are not a union of well-separated clusters (either in the original space or feature space of the kernel) Little et al. (2020), which was the motivation for the initial development of DBSCAN. There exist DBSCAN-like spectral clustering algorithms that are robust to outliers by using path-based similarity Chang and Yeung (2008); Little et al. (2020), but these algorithms have no bias towards QLCs or other degenerate clusters. For these reasons, we do not include explicit comparions to these methods in this manuscript and restrict ourselves to DBSCAN/OPTICS based algorithms.

## 3 New Algorithm: LINSCAN

### 3.1 The Embedding and Distance

LINSCAN seeks to keep the advantages of DBSCAN while being applicable to the task of distinguishing QLCs. To do this, we embed data points into $\mathbb{P}(\mathbb{R}^d)$, the space of probability measures on $\mathbb{R}^d$, and then cluster the data using a notion of distance between distributions.

LINSCAN has 3 required parameters minPts, eccPts, and $\xi$ and one optional parameter $\epsilon$. minPts, $\xi$, and $\epsilon$ are identical to the corresponding parameters in OPTICS, but eccPts is a parameter specific to LINSCAN which determines how we form the distributions we use for clustering. Letting $R^m(x)$ be the $m$-nearest neighbors to $x$ in $X$, we define a mapping

$$x \in X \mapsto \mathcal{N}\left(\mu_{R^{\mathrm{eccPts}}(x)}, \frac{\Sigma_{R^{\mathrm{eccPts}}(x)}}{\left\|\Sigma_{R^{\mathrm{eccPts}}(x)}\right\|_2}\right)$$

Thus, we embed each point in the dataset as the normal distribution best approximating its eccPts-nearest neighbors, which allows us to cluster the points based on the local covariance of the data. Note that we rescale the covariance matrix to have maximal eigenvalue of 1.

To perform clustering in this space, we define a distance function as

$$D(P,Q) = \frac{1}{2}\left\|\Sigma_Q^{-1/2}\Sigma_P\Sigma_Q^{-1/2} - I\right\|_F + \frac{1}{2}\left\|\Sigma_P^{-1/2}\Sigma_Q\Sigma_P^{-1/2} - I\right\|_F + \frac{1}{\sqrt{2}}\|\mu_P - \mu_Q\|_{\Sigma_Q^{-1}} + \frac{1}{\sqrt{2}}\|\mu_P - \mu_Q\|_{\Sigma_P^{-1}}$$

where $P = \mathcal{N}(\mu_P, \Sigma_P)$ and $Q = \mathcal{N}(\mu_Q, \Sigma_Q)$ for positive definite $\Sigma_P$ and $\Sigma_Q$. Note that this function is symmetric and $D(P,Q) = 0$ if and only if $P = Q$. Although $D$ does not satisfy the triangle inequality and is thus not a metric, later we will discuss an approximate form of the triangle inequality that $D$ does satisfy (see Theorem 3.2). Note that by choosing to normalize the covariances as above, we have

$$D(P,Q) \geq \sqrt{2}\|\mu_P - \mu_Q\|_2 \tag{1}$$

Thus, points can be efficiently disqualified from consideration without having to calculate the more expensive matrix terms if the means are sufficiently far apart, which can be used to improve the run-time of the algorithm. This is, in particular, how we utilize $\epsilon$, as this means we can filter out pairs points using standard spatial methods (KD-Trees, etc.) in Euclidean space to filter out points that are sufficiently far apart without having to compute their distance in our distance measure.

Once the points have been embedded as distributions, we run OPTICS on $\mathcal{P} = \{P_i\}_{i=1}^m$ with Euclidean distance replaced by $D(\cdot, \cdot)$, and cluster $X$ based on the results. The full process is described in Algorithm 4 (see supplemental document).

### 3.2 Motivating the Definition of $D$

We recall that on a probability space $\mathcal{X}$, the Kullback-Leibler Divergence between two Gaussians $P = \mathcal{N}(\mu_P, \Sigma_P)$ and $Q = \mathcal{N}(\mu_Q, \Sigma_Q)$ satisfies

$$KL(P|Q) = \frac{1}{2}\log\frac{|\Sigma_Q|}{|\Sigma_P|} + \frac{1}{2}\mathrm{tr}(\Sigma_Q^{-1}\Sigma_P - I) + \frac{1}{2}(\mu_P - \mu_Q)^T\Sigma_Q^{-1}(\mu_P - \mu_Q)$$

One can show (see supplemental document) that if $\left\|\Sigma_Q^{-1/2}\Sigma_P\Sigma_Q^{-1/2} - I\right\|_F < 1$, then

$$KL(P|Q) = \frac{1}{4}\left\|\Sigma_Q^{-1/2}\Sigma_P\Sigma_Q^{-1/2} - I\right\|_F^2 + o\left(\mathrm{tr}\left(\left(\Sigma_Q^{-1/2}\Sigma_P\Sigma_Q^{-1/2} - I\right)^3\right)\right) + \frac{1}{2}(\mu_P - \mu_Q)^T\Sigma_Q^{-1}(\mu_P - \mu_Q)$$

So, we can define an approximation of $KL(P|Q)$ by

$$M(P|Q) = \frac{1}{4}\left\|\Sigma_Q^{-1/2}\Sigma_P\Sigma_Q^{-1/2} - I\right\|_F^2 + \frac{1}{2}(\mu_P - \mu_Q)^T\Sigma_Q^{-1}(\mu_P - \mu_Q)$$

This motivates the symmetric distance function $D(P,Q)$, which takes term-wise square roots of $M(P|Q)$ and $M(Q|P)$ to more closely approximate a metric.

We note that other metrics, in particular Wasserstein-2 distance, also have a closed form between Gaussians. While this is a metric, the distance between the means and covariances are independent, whereas $D$ incorporates the Mahalanobis distance and penalizes differences in mean more heavily in directions orthogonal to the local linearity of the point. Furthermore, the Wasserstein-2 distance scales polynomial in the magnitude of the eigenvalues of the covariance matrices as the angles diverge, whereas $D$ penalizes orthogonal covariance inversely to the size of the minimum eigenvalues for high eccentricity clusters. This ensures that two points with large deviations in covariance direction will be far apart in $D$, even if spatially close to one another, and thus these points will not fall into the same cluster.

### 3.3  Behavior of $D$

Our distance measure is not a metric. However, in the case of Gaussians, it satisfies the properties of symmetry and separation of points in general, and, as we will show in Theorem 3.2, it satisfies a relaxed form of the triangle inequality.

**Lemma 3.1** *Let $P = \mathcal{N}(\mu_P, \Sigma_P)$ and $Q = \mathcal{N}(\mu_Q, \Sigma_Q)$ be Gaussians. Then,*

  *a. $D$ is symmetric, meaning*
$$D(P,Q) = D(Q,P)$$

  *b. $D(P,Q) = 0$ iff $P = Q$ (in particular $D(P,P) = 0$)*

Proof:

  a. Trivial

  b. Note that by the definition of $D$,

$$D(P,Q) = 0 \iff \|\mu_P - \mu_Q\|_{\Sigma_Q^{-1}} = \|\mu_P - \mu_Q\|_{\Sigma_P^{-1}} = 0 \text{ and } \Sigma_Q^{-1/2}\Sigma_P\Sigma_Q^{-1/2} = \Sigma_P^{-1/2}\Sigma_Q\Sigma_P^{-1/2} = I$$

   Since we assume all of the covariance matrices are invertible, the first equalities hold iff $\mu_P = \mu_Q$. Similarly, the second equalities hold iff $\Sigma_P = \Sigma_Q$. Hence, $D(P,Q) = 0$ iff $P = Q$

While $D$ does not satisfy the full triangle inequality, one can show that it satisfies a slightly relaxed version. We utilize the matrix commutator $[\cdot,\cdot] : \mathbb{R}^{d \times d} \times \mathbb{R}^{d \times d} \to \mathbb{R}^{d \times d}$, which measures the degree to which two matrices commute via $[A,B] := AB - BA$.

**Theorem 3.2** *Let $\epsilon > 0$. If $D(P,Q), D(Q,K) \leq \epsilon$, then*
$$D(P,K) \leq D(P,Q) + D(Q,K) + \sqrt{2}\epsilon + \sqrt{2}\epsilon\sqrt{1+\epsilon} + \epsilon^2 + E(P,Q,K),$$
*where $E(P,Q,K) = 0$ if $\Sigma_P$, $\Sigma_Q$, and $\Sigma_K$ commute and otherwise has a (loose) bound of*

$$E(P,Q,K) \leq C_{Q,K}\left\|\left[\Sigma_P, \Sigma_Q^{-1/2}\right]\right\|_F + C_{P,Q}\left\|\left[\Sigma_K, \Sigma_Q^{-1/2}\right]\right\|_F$$
$$+ C'_{Q,K}\left\|\left[\Sigma_K^{-1/2}, \Sigma_Q^{-1/2}\Sigma_P\Sigma_Q^{-1/2}\right]\right\|_F + C'_{P,Q}\left\|\left[\Sigma_P^{-1/2}, \Sigma_Q^{-1/2}\Sigma_K\Sigma_Q^{-1/2}\right]\right\|_F,$$

*and each constant $C_{i,j}$ depends on ratios of eigenvalues of $\Sigma_i$ and $\Sigma_j$ for $i, j \in \{P, Q, R\}$.*

The proof relies on a significant number of inequalities and is provided in the appendix. The proof proceeds by separating the first two terms of $D(P, K)$ from the last two and showing that each pair individually satisfies the triangle inequality with small additive errors.

Importantly, this shows that for small values of $\epsilon$, $D$ behaves approximately like a metric, which allows us to bound the diameter of any cluster in terms of $\epsilon$ and the number of steps between points in the cluster. This ensures that points whose local neighborhoods are nearly orthogonal are not clustered together.

Compare this to the best results proven previously for the approximate triangle inequality of the unmodified KL-Divergence between Gaussians in Zhang et al. (2021), which was of exponential order.

## 4 Numerical Results

Experiments with synthetic data sets revealed that some clusters identified by LINSCAN may not appear as sufficiently "linear" upon visual inspection (e.g., due to high scatter of data points). Therefore we introduce an additional quality check whereby we compute the covariance matrix of each cluster. In the case of $\mathbb{R}^2$, we set a minimal threshold $\tau$ on the ratio of the minimum eigenvalue to the maximum eigenvalue of the covariance matrix and remove the groups that do not meet this threshold.

### 4.1 Runtime Comparisons

One possible issue with working with a custom distance measure is the cost of calculating all possible distances. In figure 2 we plot the cost of calculating all pairwise distances for datasets of various sizes as eccPts varies, as well as on a system with a GPU to accelerate the distance computation and one without. We can see that even for large amounts of points, the runtime for calculating both distance measures across all pairs of points is less than a second on average. For comparison, the runtime of the actual clustering algorithm is on the order of 20 seconds on our machine, so although the runtime is higher for LINSCAN's distance measure, that cost is dwarfed by the core clustering algorithm.

On top of this, if further speedups are required, we can use out-of-the box spatial indexing methods. Using 1, we can lower bound the distance between two distributions by the distance between their means, which means that we can perform an efficient initial step where we filter out pairs whose means are sufficiently far apart before calculating our distance measure on the remaining pairs.

Figure 2: Distance Runtime

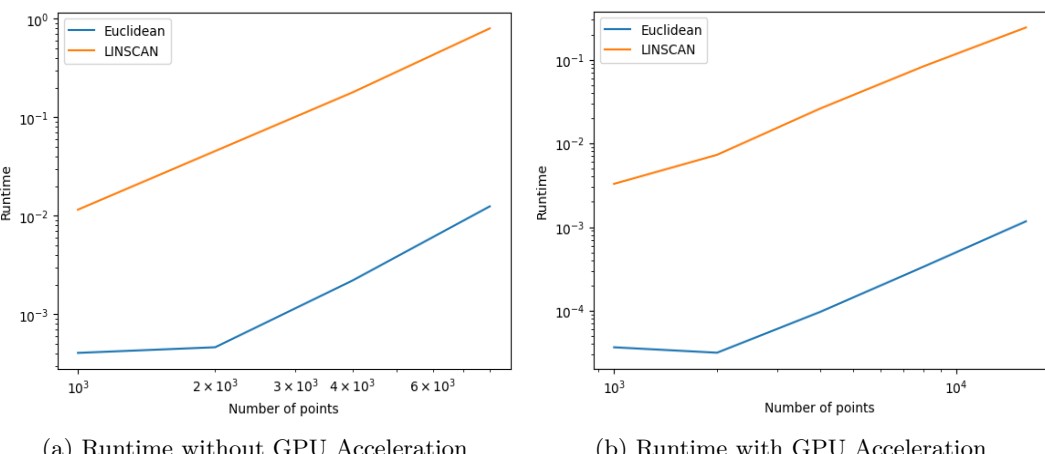

(a) Runtime without GPU Acceleration       (b) Runtime with GPU Acceleration

## 4.2 Example Datasets

Figure 3 shows an example of synthetic data with two QLCs intersecting at a high angle. Note that unlike DBSCAN, LINSCAN is able to separate the two QLCs. Further, we can see the dependence on point order in ADCN, as the clustering behavior is sensitive to initialization, in contrast to LINSCAN which is fully deterministic and independent of the ordering of points.

Figure 4a shows the results of applying LINSCAN to the same data as in Figure 1, Figure 4b shows the clusters with the noise points removed, and Figure 4c shows the results of removing clusters with spectral ratio greater than $\frac{1}{2}$. Note the separation of clusters in the bottom left and top left corners in comparison to the results from DBSCAN.

Figure 5 shows the results of applying LINSCAN to real data representing earthquake epicenters in Southern California (Fialko and Jin, 2021). Not only does LINSCAN identify QLCs and remove the "diffuse" background seismicity, but it is also able to identify the clusters at multiple distinct scales by varying minPts. If we try to do the same thing with OPTICS we get multiple clusters, but we fail to form specifically linear clusters. We don't contrast against ADCN, as getting a representative picture of its performance on a dataset of this size requires applying the algorithm many times from different initializations due to the dependence of ADCN of the order of the points.

Figure 3: Crossing Lines

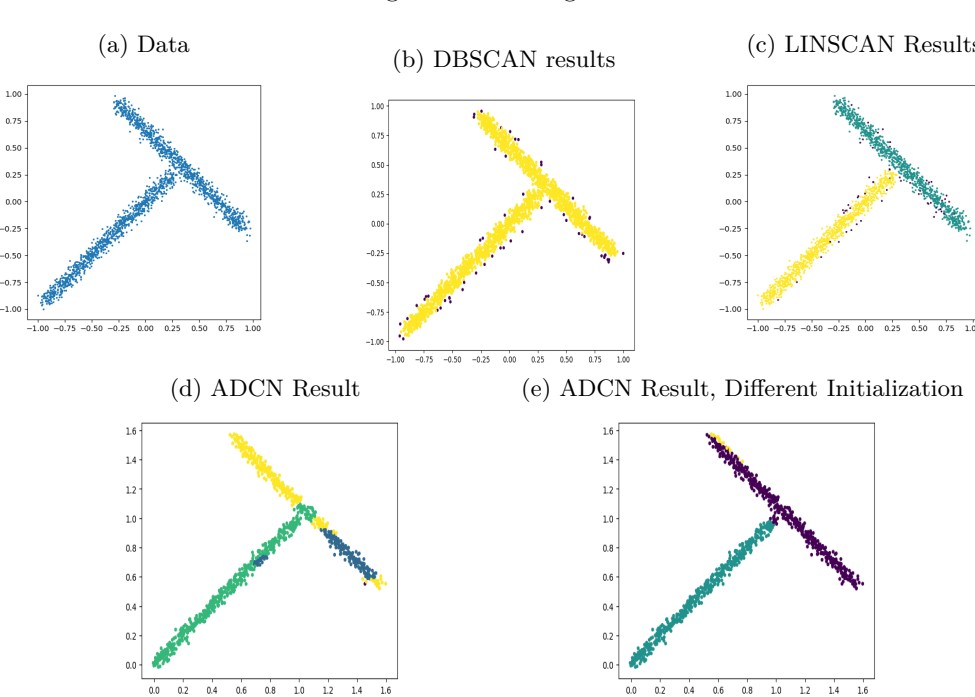

(a) Data     (b) DBSCAN results     (c) LINSCAN Results

(d) ADCN Result     (e) ADCN Result, Different Initialization

## 4.3 Measuring Performance

To quantitatively evaluate the algorithm performance, we conducted several tests on synthetic, labeled data. We generated 10 linear clusters, 5 isotropic clusters, and 10 pairs of linear clusters intersecting at angles in the range $[.1\pi, .9\pi]$ and separated them from one another in space. An example is given in Figure 6a.

To score the performance, we use the Adjusted Rand Index from Hubert and Arabie (1985):

Figure 4: Synthetic Data

(a) LINSCAN Results      (b) LINSCAN with Noise Re-   (c) LINSCAN with Isotropic
                         moved                        Clusters Removed

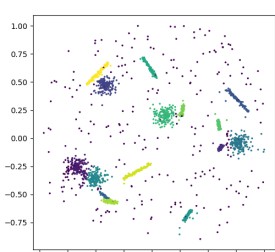 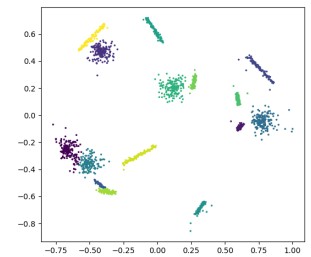 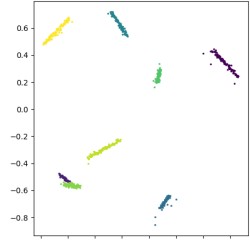

**Definition 4.1 (Rand Index)** *Let $X = \{x_1, ..., x_n\}$ be a set of elements and consider two partitions $\mathcal{C} = \{C_1, ..., C_m\}$ and $\mathcal{C}' = \{C'_1, ..., C'_n\}$, i.e. $C_i \subseteq X$ and $C'_i \subseteq X$ for all $i$ and*

$$X = \bigcup_{i=1}^{m} C_i = \bigcup_{i=1}^{n} C'_i$$

*with*

$$C_i \cap C_j = C'_i \cap C'_j = \emptyset$$

*for all $i \neq j$. Let*

$$a := \# \left\{ (x,y) \in X \times X : x \neq y, \exists i, j \ s.t. \ x, y \in C_i, x, y \in C'_j \right\}$$

*and*

$$b := \# \{ (x,y) \in X \times X : x \neq y, \exists i, j, k, l \ s.t. \ i \neq j, k \neq l, x \in C_i, x \in C'_k, y \in C_j, y \in C'_l \}$$

*$a$ is the number of pairs of elements of $X$ such that both elements are in the same cluster in $\mathcal{C}$ and $\mathcal{C}'$ and $b$ is the number of pairs of elements of $X$ such that both elements are in different clusters in both $\mathcal{C}$ and $\mathcal{C}'$. Then, the Rand Index is given by*

$$R(X, \mathcal{C}, \mathcal{C}') = \frac{a+b}{\binom{n}{2}}$$

*So, $R(\mathcal{C}, \mathcal{C}')$ is the fraction of pairs of elements of $X$ such that $\mathcal{C}$ and $\mathcal{C}'$ both agree about whether the pair of elements lie in the same cluster or not. Note that $R$ is symmetric in $\mathcal{C}$ and $\mathcal{C}'$ and lies in the interval $[0,1]$. However, random partitions are not guaranteed to have near-zero pairwise Rand Index. To remedy this, we use the Adjusted Rand Index*

$$ARI(\mathcal{C}, \mathcal{C}') = \frac{R(\mathcal{C}, \mathcal{C}') - \mathbb{E}\left[R(\mathcal{C}, \mathcal{C}')\right]}{1 - \mathbb{E}\left[R(\mathcal{C}, \mathcal{C}')\right]}$$

*where the expectation is taken over random partitions of $X$ with the same number of clusters and number of elements in each cluster as $\mathcal{C}$ and $\mathcal{C}'$. Unlike the Rand Index, the Adjusted Rand Index may be negative, but it is a better measure of the similarity between two partitions as the Rand Index tends to be higher on average for finer partitions regardless of similarity.*

Figure 5: Real Data

(a) Data

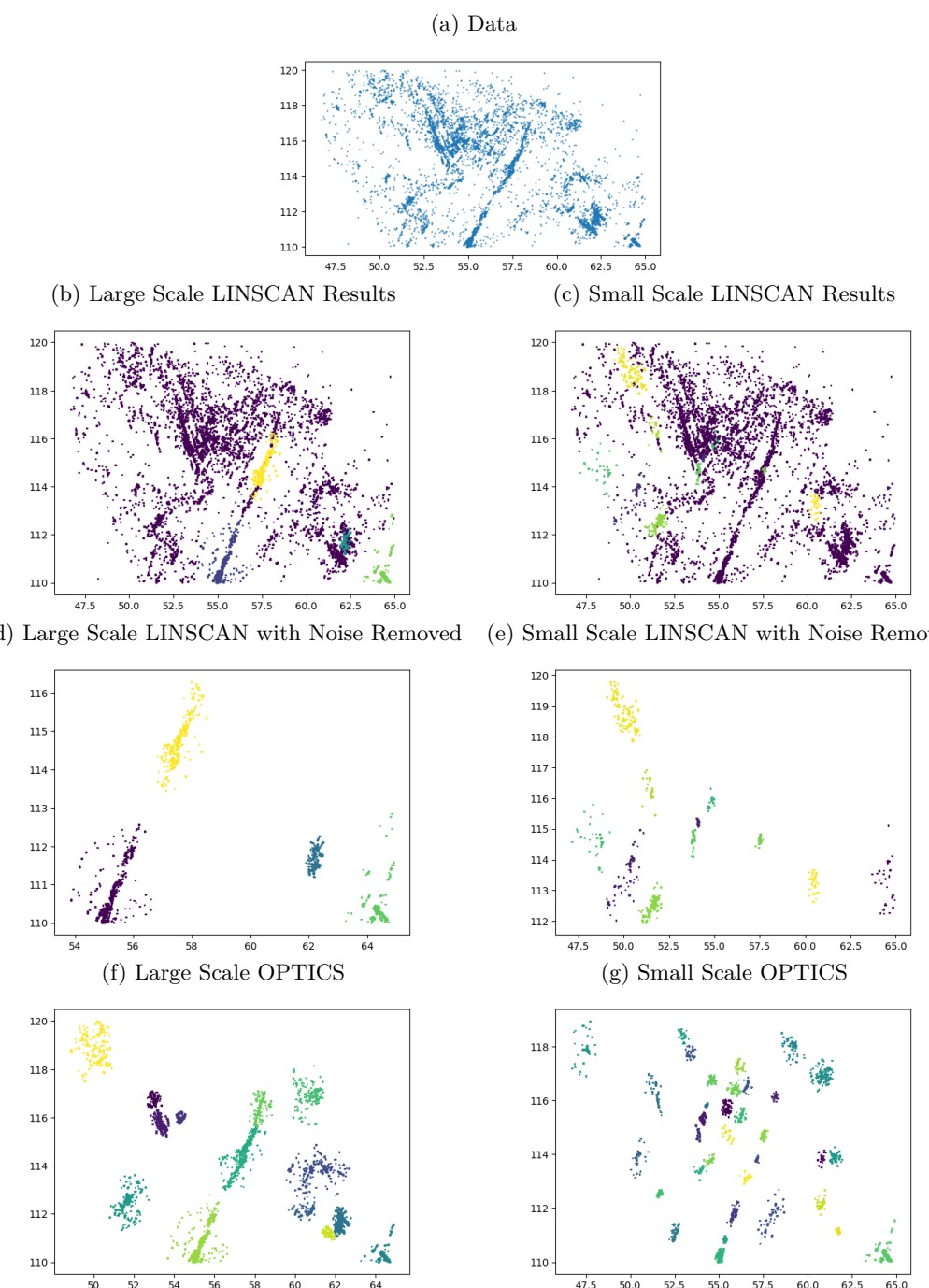

(b) Large Scale LINSCAN Results · (c) Small Scale LINSCAN Results

(d) Large Scale LINSCAN with Noise Removed · (e) Small Scale LINSCAN with Noise Removed

(f) Large Scale OPTICS · (g) Small Scale OPTICS

## 5  Experimental results

In our synthetic experiments, we perform hyperparameter optimization of both LINSCAN and OP-TICS (for comparison) on 10 synthetic datasets using a random search of the feature space for 500

trials, applying our spectral filtering to both LINSCAN and OPTICS. We then report the test accuracy of both algorithms on 40 synthetic datasets. The results are as follows:

| Algorithm | OPTICS | LINSCAN |
|---|---|---|
| Validation ARI | 46.73% | 61.48% |
| Testing ARI | 46.40% | 64.19% |

In particular, even though the parameter space for LINSCAN is much larger than OPTICS (optimizing minPts, eccPts, $\xi$, and $\tau$ compared to just minPts, $\epsilon$, and $\tau$), LINSCAN performed better on both the validation data and the testing data and generalized as well as or better than OPTICS. A sample of the performance of LINSCAN and OPTICS on generated data is given in Figure 6.

Figure 6: Generated Data

(a) Dataset            (b) LINSCAN Results            (c) OPTICS Results

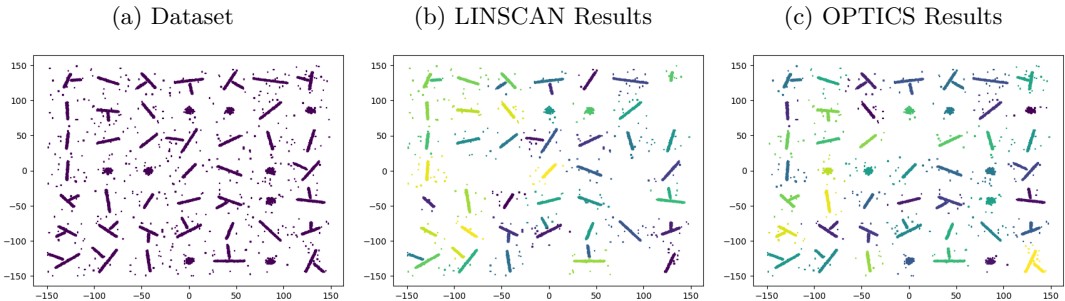

**Code availability**

The source codes are available for downloading at the link, which has been anonymized for the purposes of blind review: `http://bit.ly/419AFZg`

## 6  Conclusion

We present a method for detecting linear clusters in noisy data. This is done using a novel distance measure, motivated by KL-divergence between small data-driven Gaussian representations of the points, inside of the OPTICS algorithm. We also prove that our distance measure has more regular local behavior than the standard symmetrized KL Divergence. This approach significantly outperforms the DBSCAN family of algorithms that do not have a priori bias towards lineated clusters. Finally, we have shown our approach is shown to be effective in detecting linear slip faults in seismic data and are currently exploring additional applications of our algorithm in other domains.

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

# A   Approximation of $KL(P|Q)$

First, $\log|A|$ is the logarithm of the product of the eigenvalues of $A$, which is the same as the sum of the logarithms of the eigenvalues. Therefore,

$$\log|A| = \text{tr}(\log(A))$$

where $\log(A)$ is the matrix logarithm, which exists and is unique for any positive definite matrix $A$. In particular, if $A = Q\Lambda Q^T$ for orthogonal $Q$ and diagonal $\Lambda \succ 0$,

$$\log A = Q\log(\Lambda)Q^T$$

where $\log\Lambda$ is the diagonal matrix given by applying the logarithm entrywise to each diagonal entry. Given this,

$$\log\frac{|\Sigma_Q|}{|\Sigma_P|} = \log|\Sigma_Q| - \log|\Sigma_P|$$
$$= \text{tr}\left(\log(\Sigma_Q) - \log(\Sigma_P)\right)$$

Next, for any positive definite matrices $A$ and $B$,

$$\text{tr}\left(\log(AB)\right) = \text{tr}(\log(A)) + \text{tr}(\log(B))$$
$$\log(A^{-1}) = -\log(A)$$

Furthermore, if $\|A - I\|_F < 1$, then the sum

$$\sum_{n=1}^{\infty}(-1)^{k+1}\frac{(A-I)^k}{k}$$

converges in $\|\cdot\|_F$ to $\log(A)$. Combining all of this, if $\left\|\Sigma_Q^{-1/2}\Sigma_P\Sigma_Q^{-1/2} - I\right\| < 1$ then

$$\text{tr}\left(\log(\Sigma_Q) - \log(\Sigma_P)\right)$$
$$= -\text{tr}\left(\log\left(\Sigma_Q^{-1/2}\right) + \log\left(\Sigma_P\right) + \log\left(\Sigma_Q^{-1/2}\right)\right)$$
$$= -\text{tr}\left(\log\left(\Sigma_Q^{-1/2}\Sigma_P\Sigma_Q^{-1/2}\right)\right)$$
$$= -\text{tr}\left(\sum_{k=1}^{\infty}(-1)^{k+1}\frac{\left(\Sigma_Q^{-1/2}\Sigma_P\Sigma_Q^{-1/2} - I\right)^k}{k}\right)$$
$$= -\sum_{k=1}^{\infty}(-1)^{k+1}\frac{\text{tr}\left(\left(\Sigma_Q^{-1/2}\Sigma_P\Sigma_Q^{-1/2} - I\right)^k\right)}{k}$$
$$= -\text{tr}\left(\Sigma_Q^{-1/2}\Sigma_P\Sigma_Q^{-1/2} - I\right) + \frac{1}{2}\text{tr}\left(\left(\Sigma_Q^{-1/2}\Sigma_P\Sigma_Q^{-1/2} - I\right)^2\right) + o\left(\text{tr}\left(\left(\Sigma_q^{-1/2}\Sigma_P\Sigma_Q^{-1/2} - I\right)^3\right)\right)$$
$$= -\text{tr}\left(\Sigma_Q^{-1/2}\Sigma_P\Sigma_Q^{-1/2} - I\right) + \frac{1}{2}\left\|\Sigma_Q^{-1/2}\Sigma_P\Sigma_Q^{-1/2} - I\right\|_F^2 + o\left(\text{tr}\left(\left(\Sigma_Q^{-1/2}\Sigma_P\Sigma_Q^{-1/2} - I\right)^3\right)\right)$$

where in the last line we used the fact that $\Sigma_Q^{-1/2}\Sigma_P\Sigma_Q^{-1/2} - I$ is symmetric and for any symmetric matrix $A$

$$\text{tr}(A^2) = \text{tr}(A^T A) = \|A\|_F^2$$

Next, note that

$$\mathrm{tr}(\Sigma_Q^{-1}\Sigma_P - I) = \mathrm{tr}(\Sigma_Q^{-1/2}\Sigma_P\Sigma_Q^{-1/2} - I)$$

So, combined with the prior derivations,

$$\frac{1}{2}\log\frac{|\Sigma_Q|}{|\Sigma_P|} + \frac{1}{2}\mathrm{tr}(\Sigma_Q^{-1}\Sigma_P - I)$$
$$= \frac{1}{4}\left\|\Sigma_Q^{-1/2}\Sigma_P\Sigma_Q^{-1/2} - I\right\|_F^2 + o\left(\mathrm{tr}\left(\left(\Sigma_Q^{-1/2}\Sigma_P\Sigma_Q^{-1/2} - I\right)^3\right)\right)$$

from which the rest of the approximation follows.

## B   Proof of Relaxed Triangle Inequality

We recall that

$$D(P,Q) = \frac{1}{2}\left\|\Sigma_Q^{-1/2}\Sigma_P\Sigma_Q^{-1/2} - I\right\|_F + \frac{1}{2}\left\|\Sigma_P^{-1/2}\Sigma_Q\Sigma_P^{-1/2} - I\right\|_F$$
$$+ \frac{1}{\sqrt{2}}\|\mu_P - \mu_Q\|_{\Sigma_Q^{-1}} + \frac{1}{\sqrt{2}}\|\mu_P - \mu_Q\|_{\Sigma_P^{-1}}$$

These terms are all nonnegative, so if $D(P,Q) \leq \epsilon$ then each term is at most $\epsilon$. To show the relaxed triangle inequality, we define

$$D_1(P,Q) := \left\|\Sigma_Q^{-1/2}\Sigma_P\Sigma_Q^{-1/2} - I\right\|_F + \left\|\Sigma_P^{-1/2}\Sigma_Q\Sigma_P^{-1/2} - I\right\|_F$$

and

$$D_2(P,Q) := \|\mu_P - \mu_Q\|_{\Sigma_Q^{-1}} + \|\mu_P - \mu_Q\|_{\Sigma_P^{-1}}$$

so that

$$D(P,Q) = \frac{1}{2}D_1(P,Q) + \frac{1}{\sqrt{2}}D_2(P,Q)$$

Then,

$$D_2(P,K) = \|\mu_P - \mu_K\|_{\Sigma_K^{-1}} + \|\mu_P - \mu_K\|_{\Sigma_P^{-1}}$$
$$\leq \|\mu_P - \mu_Q\|_{\Sigma_K^{-1}} + \|\mu_Q - \mu_K\|_{\Sigma_K^{-1}} + \|\mu_P - \mu_Q\|_{\Sigma_P^{-1}} + \|\mu_Q - \mu_K\|_{\Sigma_P^{-1}}$$
$$= D_2(P,Q) + D_2(Q,K) + \|\mu_P - \mu_Q\|_{\Sigma_K^{-1}} - \|\mu_P - \mu_Q\|_{\Sigma_Q^{-1}} + \|\mu_Q - \mu_K\|_{\Sigma_P^{-1}} - \|\mu_Q - \mu_K\|_{\Sigma_Q^{-1}}$$

Note that

$$\|\mu_P - \mu_Q\|_{\Sigma_K^{-1}} - \|\mu_P - \mu_Q\|_{\Sigma_Q^{-1}} = \left\|\Sigma_K^{-1/2}(\mu_P - \mu_Q)\right\|_2 - \left\|\Sigma_Q^{-1/2}(\mu_P - \mu_Q)\right\|_2$$
$$\leq \left\|\Sigma_K^{-1/2}(\mu_P - \mu_Q) - \Sigma_Q^{-1/2}(\mu_P - \mu_Q)\right\|_2$$
$$= \left\|\left(\Sigma_K^{-1/2} - \Sigma_Q^{-1/2}\right)(\mu_P - \mu_Q)\right\|_2$$
$$= \left\|\left(\Sigma_K^{-1/2}\Sigma_Q^{1/2} - I\right)\Sigma_Q^{-1/2}(\mu_P - \mu_Q)\right\|_2$$
$$\leq \left\|\Sigma_K^{-1/2}\Sigma_Q^{1/2} - I\right\|_2\left\|\Sigma_Q^{-1/2}(\mu_P - \mu_Q)\right\|_2$$
$$= \left\|\Sigma_K^{-1/2}\Sigma_Q^{1/2} - I\right\|_2\|\mu_P - \mu_Q\|_{\Sigma_Q^{-1}}$$
$$\leq \left\|\Sigma_K^{-1/2}\Sigma_Q^{1/2} - I\right\|_2\epsilon$$

Now, note that $\left\|\Sigma_K^{-1/2}\Sigma_Q^{1/2} - I\right\|_2$ is the square root of the maximal eigenvalue of

$$(\Sigma_K^{-1/2}\Sigma_Q^{1/2} - I)^T(\Sigma_K^{-1/2}\Sigma_Q^{1/2} - I)$$

Therefore,

$$
\begin{aligned}
\left\|\Sigma_K^{-1/2}\Sigma_Q^{1/2} - I\right\|_2^2 &= \left\|(\Sigma_K^{-1/2}\Sigma_Q^{1/2} - I)^T(\Sigma_K^{-1/2}\Sigma_Q^{1/2} - I)\right\|_2 \\
&= \left\|\Sigma_K^{-1/2}\Sigma_Q\Sigma_K^{-1/2} - \Sigma_K^{-1/2}\Sigma_Q^{1/2} - \Sigma_Q^{1/2}\Sigma_K^{-1/2} + I\right\|_2 \\
&\leq \left\|\Sigma_K^{-1/2}\Sigma_Q\Sigma_K^{-1/2} - I\right\|_2 + \left\|2I - \Sigma_K^{-1/2}\Sigma_Q^{1/2} - \Sigma_Q^{1/2}\Sigma_K^{-1/2}\right\|_2 \\
&\leq \left\|\Sigma_K^{-1/2}\Sigma_Q\Sigma_K^{-1/2} - I\right\|_2 + \left\|I - \Sigma_K^{-1/2}\Sigma_Q^{1/2}\right\|_2 + \left\|I - \Sigma_Q^{1/2}\Sigma_K^{-1/2}\right\|_2 \\
&= \left\|\Sigma_K^{-1/2}\Sigma_Q\Sigma_K^{-1/2} - I\right\|_2 + 2\left\|I - \Sigma_K^{-1/2}\Sigma_Q^{1/2}\right\|_2 \\
&= \left\|\Sigma_K^{-1/2}\Sigma_Q\Sigma_K^{-1/2} - I\right\|_2 + 2\left\|\Sigma_K^{-1/2}\Sigma_Q^{1/2} - I\right\|_2
\end{aligned}
$$

Solving this for $\left\|\Sigma_K^{-1/2}\Sigma_Q^{1/2} - I\right\|_2$, we get

$$\left\|\Sigma_K^{-1/2}\Sigma_Q^{1/2} - I\right\|_2 \leq 1 + \sqrt{1 + \left\|\Sigma_K^{-1/2}\Sigma_Q\Sigma_K^{-1/2} - I\right\|_2} \leq 1 + \sqrt{1 + \left\|\Sigma_K^{-1/2}\Sigma_Q\Sigma_K^{-1/2} - I\right\|_F} \leq 1 + \sqrt{1 + \epsilon}$$

So,

$$\|\mu_P - \mu_Q\|_{\Sigma_K^{-1}} - \|\mu_P - \mu_Q\|_{\Sigma_Q^{-1}} \leq \left\|\Sigma_K^{-1/2}\Sigma_Q^{1/2} - I\right\|_2 \epsilon \leq \epsilon + \epsilon\sqrt{1 + \epsilon}$$

A similar statement holds for $\|\mu_Q - \mu_K\|_{\Sigma_P^{-1}} - \|\mu_Q - \mu_K\|_{\Sigma_Q^{-1}}$, so

$$D_2(P, K) \leq D_2(P, Q) + D_2(Q, K) + 2\epsilon + 2\epsilon\sqrt{1 + \epsilon}$$

Next,

$$
\begin{aligned}
&\left\|\Sigma_P^{-1/2}\Sigma_K\Sigma_P^{-1/2} - I\right\|_F - \left\|\Sigma_Q^{-1/2}\Sigma_K\Sigma_Q^{-1/2} - I\right\|_F - \left\|\Sigma_P^{-1/2}\Sigma_Q\Sigma_P^{-1/2} - I\right\|_F \\
&\leq \left\|\Sigma_P^{-1/2}\Sigma_K\Sigma_P^{-1/2} - \Sigma_Q^{-1/2}\Sigma_K\Sigma_Q^{-1/2}\right\|_F - \left\|\Sigma_P^{-1/2}\Sigma_Q\Sigma_P^{-1/2} - I\right\|_F \\
&\leq \left\|\Sigma_P^{-1/2}\Sigma_K\Sigma_P^{-1/2} - \Sigma_Q^{-1/2}\Sigma_K\Sigma_Q^{-1/2} - \Sigma_P^{-1/2}\Sigma_Q\Sigma_P^{-1/2} + I\right\|_F \\
&= \left\|\left(I - \Sigma_Q^{-1/2}\Sigma_P\Sigma_Q^{-1/2}\right)\left(I - \Sigma_K^{-1/2}\Sigma_Q\Sigma_K^{-1/2}\right) + \Sigma_K^{-1/2}\Sigma_P\Sigma_K^{-1/2} - \Sigma_Q^{-1/2}\Sigma_P\Sigma_Q^{-1/2}\Sigma_K^{-1/2}\Sigma_Q\Sigma_K^{-1/2}\right\|_F \\
&\leq \left\|I - \Sigma_Q^{-1/2}\Sigma_P\Sigma_Q^{-1/2}\right\|_F\left\|I - \Sigma_K^{-1/2}\Sigma_Q\Sigma_K^{-1/2}\right\|_F + \left\|\Sigma_K^{-1/2}\Sigma_P\Sigma_K^{-1/2} - \Sigma_Q^{-1/2}\Sigma_P\Sigma_Q^{-1/2}\Sigma_K^{-1/2}\Sigma_Q\Sigma_K^{-1/2}\right\|_F \\
&\leq \epsilon^2 + \left\|\Sigma_K^{-1/2}\Sigma_P\Sigma_K^{-1/2} - \Sigma_Q^{-1/2}\Sigma_P\Sigma_Q^{-1/2}\Sigma_K^{-1/2}\Sigma_Q\Sigma_K^{-1/2}\right\|_F
\end{aligned}
$$

A similar argument shows

$$
\begin{aligned}
&\left\|\Sigma_K^{-1/2}\Sigma_P\Sigma_K^{-1/2} - I\right\|_F - \left\|\Sigma_Q^{-1/2}\Sigma_P\Sigma_Q^{-1/2} - I\right\|_F - \left\|\Sigma_K^{-1/2}\Sigma_Q\Sigma_K^{-1/2} - I\right\|_F \\
&\leq \epsilon^2 + \left\|\Sigma_P^{-1/2}\Sigma_K\Sigma_P^{-1/2} - \Sigma_Q^{-1/2}\Sigma_K\Sigma_Q^{-1/2}\Sigma_P^{-1/2}\Sigma_Q\Sigma_P^{-1/2}\right\|_F
\end{aligned}
$$

Combining these,

$$D_1(P, K) \le D_1(P, Q) + D_1(Q, K) + 2\epsilon^2$$
$$+ \left\| \Sigma_K^{-1/2} \Sigma_P \Sigma_K^{-1/2} - \Sigma_Q^{-1/2} \Sigma_P \Sigma_Q^{-1/2} \Sigma_K^{-1/2} \Sigma_Q \Sigma_K^{-1/2} \right\|_F$$
$$+ \left\| \Sigma_P^{-1/2} \Sigma_K \Sigma_P^{-1/2} - \Sigma_Q^{-1/2} \Sigma_K \Sigma_Q^{-1/2} \Sigma_P^{-1/2} \Sigma_Q \Sigma_P^{-1/2} \right\|_F$$

If $[A, B] = AB - BA$ is the commutator of $A$ and $B$,

$$\left\| \Sigma_K^{-1/2} \Sigma_P \Sigma_K^{-1/2} - \Sigma_Q^{-1/2} \Sigma_P \Sigma_Q^{-1/2} \Sigma_K^{-1/2} \Sigma_Q \Sigma_K^{-1/2} \right\|_F$$
$$\le \left\| \Sigma_K^{-1/2} \Sigma_P \Sigma_K^{-1/2} - \Sigma_K^{-1/2} \Sigma_Q^{-1/2} \Sigma_P \Sigma_Q^{-1/2} \Sigma_Q \Sigma_K^{-1/2} \right\|_F$$
$$+ \left\| \Sigma_K^{-1/2} \Sigma_Q^{-1/2} \Sigma_P \Sigma_Q^{-1/2} \Sigma_Q \Sigma_K^{-1/2} - \Sigma_Q^{-1/2} \Sigma_P \Sigma_Q^{-1/2} \Sigma_K^{-1/2} \Sigma_Q \Sigma_K^{-1/2} \right\|_F$$
$$= \left\| \Sigma_K^{-1/2} \Sigma_P \Sigma_K^{-1/2} - \Sigma_K^{-1/2} \Sigma_Q^{-1/2} \Sigma_P \Sigma_Q^{-1/2} \Sigma_Q \Sigma_K^{-1/2} \right\|_F$$
$$+ \left\| \left[ \Sigma_K^{-1/2}, \Sigma_Q^{-1/2} \Sigma_P \Sigma_Q^{-1/2} \right] \Sigma_Q \Sigma_K^{-1/2} \right\|_F$$
$$= \left\| \Sigma_K^{-1/2} \Sigma_P \Sigma_Q^{-1/2} \Sigma_Q^{-1/2} \Sigma_Q \Sigma_K^{-1/2} - \Sigma_K^{-1/2} \Sigma_Q^{-1/2} \Sigma_P \Sigma_Q^{-1/2} \Sigma_Q \Sigma_K^{-1/2} \right\|_F$$
$$+ \left\| \left[ \Sigma_K^{-1/2}, \Sigma_Q^{-1/2} \Sigma_P \Sigma_Q^{-1/2} \right] \Sigma_Q \Sigma_K^{-1/2} \right\|_F$$
$$= \left\| \Sigma_K^{-1/2} \left[ \Sigma_P, \Sigma_Q^{-1/2} \right] \Sigma_Q^{-1/2} \Sigma_Q \Sigma_K^{-1/2} \right\|_F + \left\| \left[ \Sigma_K^{-1/2}, \Sigma_Q^{-1/2} \Sigma_P \Sigma_Q^{-1/2} \right] \Sigma_Q \Sigma_K^{-1/2} \right\|_F$$
$$= \left\| \Sigma_K^{-1/2} \left[ \Sigma_P, \Sigma_Q^{-1/2} \right] \Sigma_Q^{1/2} \Sigma_K^{-1/2} \right\|_F + \left\| \left[ \Sigma_K^{-1/2}, \Sigma_Q^{-1/2} \Sigma_P \Sigma_Q^{-1/2} \right] \Sigma_Q \Sigma_K^{-1/2} \right\|_F$$

Similarly,

$$\left\| \Sigma_P^{-1/2} \Sigma_K \Sigma_P^{-1/2} - \Sigma_Q^{-1/2} \Sigma_K \Sigma_Q^{-1/2} \Sigma_P^{-1/2} \Sigma_Q \Sigma_P^{-1/2} \right\|_F$$
$$\le \left\| \Sigma_P^{-1/2} \left[ \Sigma_K, \Sigma_Q^{-1/2} \right] \Sigma_Q^{1/2} \Sigma_P^{-1/2} \right\|_F + \left\| \left[ \Sigma_P^{-1/2}, \Sigma_Q^{-1/2} \Sigma_K \Sigma_Q^{-1/2} \right] \Sigma_Q \Sigma_P^{-1/2} \right\|_F$$

So finally, if we let

$$E(P, Q, K) := \frac{1}{2} \left\| \Sigma_K^{-1/2} \left[ \Sigma_P, \Sigma_Q^{-1/2} \right] \Sigma_Q^{1/2} \Sigma_K^{-1/2} \right\|_F + \frac{1}{2} \left\| \left[ \Sigma_K^{-1/2}, \Sigma_Q^{-1/2} \Sigma_P \Sigma_Q^{-1/2} \right] \Sigma_Q \Sigma_K^{-1/2} \right\|_F$$
$$+ \frac{1}{2} \left\| \Sigma_P^{-1/2} \left[ \Sigma_K, \Sigma_Q^{-1/2} \right] \Sigma_Q^{1/2} \Sigma_P^{-1/2} \right\|_F + \frac{1}{2} \left\| \left[ \Sigma_P^{-1/2}, \Sigma_Q^{-1/2} \Sigma_K \Sigma_Q^{-1/2} \right] \Sigma_Q \Sigma_P^{-1/2} \right\|_F$$

then the theorem follows.

$E(P, Q, K)$ satisfies slow growth behaviour in our context. If $\Sigma_P, \Sigma_Q, \Sigma_K$ are jointly diagonalizable, then clearly $E(P, Q, K) = 0$ since each commutator will be 0. Beyond this, we can trivially bound $E$ by

$$E(P, Q, K) \le C_{Q,K} \left\| \left[ \Sigma_P, \Sigma_Q^{-1/2} \right] \right\|_F + C'_{Q,K} \left\| \left[ \Sigma_K^{-1/2}, \Sigma_Q^{-1/2} \Sigma_P \Sigma_Q^{-1/2} \right] \right\|_F$$
$$+ C_{P,Q} \left\| \left[ \Sigma_K, \Sigma_Q^{-1/2} \right] \right\|_F + C'_{P,Q} \left\| \left[ \Sigma_P^{-1/2}, \Sigma_Q^{-1/2} \Sigma_K \Sigma_Q^{-1/2} \right] \right\|_F,$$

and each constant $C_{i,j}$ depends on ratios of eigenvalues of $i, j \in \{P, Q, R\}$.

## C  Algorithms

---

**Algorithm 1** DBSCAN

---

**Input:** Data $X = \{x_1, ..., x_m\}$, $\epsilon > 0$, minPts $\in \mathbb{N}$
**Output:** Clusters $\{C_k\}$
$n \leftarrow 0$
$N \leftarrow \emptyset$
**while** $X \setminus (N \cup \bigcup_{k=0}^{n-1} C_k) \neq \emptyset$ **do**
  Pick $x \in X \setminus (N \cup \bigcup_{k=0}^{n-1} C_k)$
  **if** $\#R_\epsilon(x) <$ minPts **then**
    $N \leftarrow N \cup \{x\}$
  **else**
    $C_n \leftarrow \{x\}$
    $S \leftarrow R_\epsilon(x) \setminus (N \cup \{x\})$
    **while** $S \neq \emptyset$ **do**
      Pick $y \in S$
      **if** $\#R_\epsilon(y) <$ minPts **then**
        $N \leftarrow N \cup \{y\}$
        $S \leftarrow S \setminus \{y\}$
      **else**
        $C_n \leftarrow C_n \cup \{y\}$
        $S \leftarrow (S \cup R_\epsilon(y)) \setminus (N \cup C_n)$
      **end if**
    **end while**
    **if** $\#C_n <$ minPts **then**
      $N \leftarrow N \cup C_n$
      $C_n \leftarrow \emptyset$
    **else**
      $n \leftarrow n + 1$
    **end if**
  **end if**
**end while**=0

---

---

**Algorithm 2** OPTICS

---

**Input:** Data $X = \{x_1, ..., x_m\}$, $\epsilon > 0$, minPts $\in \mathbb{N}$, $n = 0$, $Q = \emptyset$
**Output:** Ordering $Q$, minimal reachability distances $d_{\min} : X \to \mathbb{R}_{\geq 0}$
**for** $p \in X$ **do**
    $d_{\mathrm{m}}(p) \leftarrow \infty$
**end for**
**for** $p \in X$ unprocessed **do**
    $N \leftarrow R_\epsilon(p)$
    Mark $p$ as processed
    $Q \leftarrow Q \cup \{p\}$
    **if** $d_{\mathrm{core}}(p) \neq \infty$ **then**
        $S = \emptyset$
        update($N, p, S, \epsilon$, minPts)
        **for** $q \in S$ **do**
            $N' \leftarrow R_\epsilon(q)$
            Mark $q$ as processed
            $Q \leftarrow Q \cup q$
            **if** $d_{\mathrm{core}}(q) \neq \infty$ **then**
                update($N, p, S, \epsilon$, minPts)
            **end if**
        **end for**
    **end if**
**end for**=0

---

**Algorithm 3** Update

---

**Input:** Neighborhood $N$, core point $p$, queue $S$, $\epsilon > 0$, minPts $\in \mathbb{N}$
**for** $o \in N$ **do**
    $d_{\mathrm{new}} = \max \{d_{\mathrm{core}}(p), \|p - o\|\}$
    **if** $d_{\min}(o) = \infty$ (Note this means $o \notin S$) **then**
        $d_{\min}(o) \leftarrow d_{\mathrm{new}}$
        $S = S \cup \{o\}$
    **else**
        **if** $d_{\mathrm{new}} < d_{\min}(o)$ **then**
            $d_{\min}(o) \leftarrow d_{\mathrm{new}}$
            Reorganize $S$ to be in increasing order by value of $d_{\min}$
        **end if**
    **end if**
**end for**=0

---

---

**Algorithm 4** LINSCAN

---

**Input:** Data $X = \{x_1, ..., x_m\}$, $\epsilon > 0$, minPts $\in \mathbb{N}$, $n = 0$, $N = \emptyset$, eccPts $\in \mathbb{N}$
**Output:** Clusters $\{C_k\}$
$n \leftarrow 0$
$N \leftarrow \emptyset$
$\mathcal{P} \leftarrow \emptyset$
**for** $x \in X$ **do**
    $\mu \leftarrow \mu_{R^{\text{eccPts}}(x)}$
    $\Sigma \leftarrow \Sigma_{R^{\text{eccPts}}(x)}$
    $P \leftarrow \mathcal{N}(\mu, \Sigma)$
    $\mathcal{P} \leftarrow \mathcal{P} \cup \{P\}$
**end for**
$\{D_k\} \leftarrow \text{OPTICS}(\mathcal{P}, \epsilon, \text{minPts})$
**for** $k \in \{0, 1, ..., n\}$ **do**
    $C_k \leftarrow \{x_i \in X : P_i \in D_k\}$
**end for**=0

---

