# OpenReview forum: "LINSCAN - A Linearity Based Clustering Algorithm"
_TMLR — Rejected by TMLR_

### Review · Reviewer_gh4L · 2025-03-01

**Summary Of Contributions:**

This paper gives a new variant of DBSCAN, called LINSCAN, motivated by identifying quasi-linear clusters (QLCs). As defined by the authors, the QLCs are clusters that are geometrically close but have different orientations.

The new algorithm LINSCAN is basically OPTICS/DBCAN, but run on a carefully designed distance measure. This new distance measure embeds the input Euclidean dataset into a distribution space, and then the distance is defined by some notion of distances between distributions, similar to KL divergence. This new distance measure is not a metric since it does not satisfy triangle inequality, but it is nonetheless still compatible with OPTICS/DBSCAN.

Experiments on small scale synthetic datasets are performed, and the proposed LINSCAN algorithm outperforms the vanilla OPTICS by a significant margin.

**Audience:**

Yes

**Broader Impact Concerns:**

None.

**Claims And Evidence:**

Yes

**Requested Changes:**

1. A formal definition for QLCs is never given. I think we need to model this rigorously.

2. Although it makes sense to consider DBSCAN/OPTICS, it is nonetheless important to also discuss related clustering methods and why they fail to address your issue. In particular, could kernel k-means be useful for QLCs?  It surely requires the parameter k, but if the model is planted, then it is possible to experiment with different kinds of and pick the correct one.

3. Section 1.2 is too vague, and unfortunately many points are not explained even in the later part of the paper. For example, bullet d, “it only fails to be one polynomially” — what do you mean by polynomially? Do we have a lemma/theorem statement of this? It’s good to ref to the place this is formally discussed. Your bullet a seems to be a known fact, and it is not proper to claim it’s your contribution (especially that it’s the first one).

4. The definition of the distance is complicated. Please elaborate the time complexity to evaluate this distance. More generally, it is good to also report the running time of your algorithms in the experiments.

5. There are obvious typos in the paper, please do another pass to improve. For example, at the end of page 6, it reads “definitionDefinition”

**Strengths And Weaknesses:**

**Strength:**

- The proposed method is mathematically sound, and the discussion for the intuition behind the new definition is helpful.

- Both theoretical analysis and experimental evaluation are provided.

- The paper is self-contained and is easy to follow.

**Weakness:**

- The new algorithm seems to be very specific to QLCs and the approach is only restricted to DBSCAN-based, which limits the impact and audience of the paper.

- The technical contribution seems to be fair, since the proposed algorithm does not introduce new steps beyond DBSCAN rather than using a new distance measure tailored to QLCs.

- The experiments are weak, since they are small scale, and are synthetic.

---

> ### Author Response · Authors · 2025-03-21
> **Response to gh4L**
>
> We thank the reviewer for their helpful comments. As noted in the official comment for the document, we are currently revising the manuscript to reflect these changes and will upload it within the 2 week response timeframe. We wanted to post responses now to ensure that these will fit with your expected changes and clarify if there are any other comments / revisions / confusions in the document.
>
> - "The new algorithm seems to be very specific to QLCs and the approach is only restricted to DBSCAN-based, which limits the impact and audience of the paper."
>
> We agree that the algorithm is tailored to this field, though we note that there have been previous considerations of learning quasi linear clusters, e.g.  “Spectral Clustering Based on Local PCA” by Arias-Castro, Lerman, and Zhang (JMLR 2017), though not in the presence of high levels of noise (e.g. non clustered points) as ours accomplishes.
>
> - "The experiments are weak, since they are small scale, and are synthetic."
>
> We have added a number of experiments and comparisons to other methods.  We have also extended the experiments on the real world dataset of earthquake locations in Southern California.
>
> - "A formal definition for QLCs is never given. I think we need to model this rigorously."
>
> Thank you for the comment.  We have added a definition of these clusters into the introduction.  Our definition has become: Quasi linear clusters are a cluster of points where: 1) each point is within $\epsilon$ of some other point in the cluster, 2) the total cluster has a nearly singular covariance matrix.
>
>
> - "Although it makes sense to consider DBSCAN/OPTICS, it is nonetheless important to also discuss related clustering methods and why they fail to address your issue. In particular, could kernel k-means be useful for QLCs? It surely requires the parameter k, but if the model is planted, then it is possible to experiment with different kinds of and pick the correct one."
>
> We have provided several comparisons across a number of experiments, in order to clarify the benefit of LINSCAN over existing algorithms.  We agree that a crafted kernel could attain similar notions of similarity to our distance measure.  However, the two largest issues with using any kernel clustering algorithm are 1) the kernel is quadratic in the number of points, which would make any eigenmap or feature space type embedding of the kernel prohibitively expensive in the number of points and make the calculation on the real world data impractical, and 2) any k-means based clustering will fail to deal with high level the noisy (i.e. non clustered) points.  We have added a discussion in the manuscript about this fact, and why this motivates the use of an augmented DBSCAN algorithm.
>
> - "Section 1.2 is too vague, and unfortunately many points are not explained even in the later part of the paper. For example, bullet d, “it only fails to be one polynomially” — what do you mean by polynomially? Do we have a lemma/theorem statement of this? It’s good to ref to the place this is formally discussed. Your bullet a seems to be a known fact, and it is not proper to claim it’s your contribution (especially that it’s the first one)."
>
> We have added exact statements in the Contributions section and citations for the claims about the poor performance of KL-divergence.  For the polynomial comment, this was us attempting to reference to our later theorem.  We have updated the language to reflect this and added proper reference to later sections.
>
> - "The definition of the distance is complicated. Please elaborate the time complexity to evaluate this distance. More generally, it is good to also report the running time of your algorithms in the experiments."
>
> We have added a set of experiments comparing runtime of our distance measure to standard Euclidean distance, and showing that the precomputing of the local covariance as well as the multi-term distance function does not add significant runtime.  We felt this was the proper comparison, as different distance measures can lead to different sizes of clusters and thus different runtimes of the downstream clustering algorithm.
>
> - "There are obvious typos in the paper, please do another pass to improve. For example, at the end of page 6, it reads “definitionDefinition”"
>
> Thank you for pointing this out.  We have fixed several typos in the manuscript, including the one mentioned here.

---

### Review · Reviewer_M2fe · 2025-03-03

**Summary Of Contributions:**

Motivated by geophysics problems, the paper studies the detection of quasi-linear clusters (QLCs) using clustering algorithms. After showing empirically that traditional clustering techniques such as DBSCAN and OPTICS are insufficient for their purposes, the authors propose an algorithm LINSCAN which extends these methods by embedding points as normal distributions that approximate their local neighborhoods and applying a distance function based on the Kullback-Leibler (KL) divergence. An approximate triangle inequality is proven and some empirical evaluation is given to compare LINSCAN with OPTICS.

**Audience:**

Yes

**Broader Impact Concerns:**

I do not foresee any broad ethics/impact concerns for this paper.

**Claims And Evidence:**

Yes

**Requested Changes:**

Given that LINSCAN requires additional computational overhead as compared to DBSCAN and OPTICS, the paper would benefit from a runtime complexity analysis of LINSCAN and empirical comparison of execution time against the prior methods. It would also benefit from an empirical evaluation standpoint to compare LINSCAN with other clustering methods (e.g. kNN, etc) and on other problem domains, besides saying "we expect there to be a number of additional applications of such an algorithm in other fields".

Minor note: The final sentence has a typo. "is other fields" should be "in other fields".

Minor note: I would recommend against saying "novel distance metric" when the proposed modification does not really satisfy triangle inequality (which metrics do).

**Strengths And Weaknesses:**

Strengths: The problem is well-motivated from a practical geophysics problem and the paper proposes an extension of a commonly used clustering method that aims to address the problem at hand. Some experiments were done to show the empirical gains of the proposed method. The proposed method also maintains invariance to point ordering, which is a very natural desirata for clustering algorithms.

Weaknesses: The algorithm seems to be very tailored to this particular seismic data problem and does not demonstrate LINSCAN's efficacy across other domains. That is, while LINSCAN demonstrates usefulness for the problem that the authors care about, I am unsure of the broader applicability of LINSCAN in other kinds of clustering problems. As an extension of OPTICS, LINSCAN also suffers from the need to tune hyperparameters.

---

> ### Author Response · Authors · 2025-03-21
> **Response to M2fe**
>
> We thank the reviewer for their helpful comments.  As noted in the official comment for the document, we are currently revising the manuscript to reflect these changes and will upload it within the 2 week response timeframe.  We wanted to post responses now to ensure that these will fit with your expected changes and clarify if there are any other comments / revisions / confusions in the document.
>
> - "Given that LINSCAN requires additional computational overhead as compared to DBSCAN and OPTICS, the paper would benefit from a runtime complexity analysis of LINSCAN and empirical comparison of execution time against the prior methods."
>
> We have added a set of experiments comparing runtime of our distance measure to standard Euclidean distance, and showing that the precomputing of the local covariance as well as the multi-term distance function does not add significant runtime.  We felt this was the proper comparison, as different distance measures can lead to different sizes of clusters and thus different runtimes of the downstream clustering algorithm.
>
> - "It would also benefit from an empirical evaluation standpoint to compare LINSCAN with other clustering methods (e.g. kNN, etc)"
>
> We have added a number of experiments and comparisons to other methods.  We have also extended the experiments on the real world dataset of earthquake locations in Southern California.
>
> - "The algorithm seems to be very tailored to this particular seismic data problem and does not demonstrate LINSCAN's efficacy across other domains. That is, while LINSCAN demonstrates usefulness for the problem that the authors care about, I am unsure of the broader applicability of LINSCAN in other kinds of clustering problems. As an extension of OPTICS, LINSCAN also suffers from the need to tune hyperparameters.  Comparison other problem domains, besides saying "we expect there to be a number of additional applications of such an algorithm in other fields".  Minor note: The final sentence has a typo. "is other fields" should be "in other fields"."
>
> We have chosen to remove that sentence and focus more on the applications within this domain.  We agree that the algorithm is tailored to this field, though we note that there have been previous considerations of learning quasi linear clusters, e.g.  “Spectral Clustering Based on Local PCA” by Arias-Castro, Lerman, and Zhang (JMLR 2017), though not in the presence of high levels of noise (e.g. non clustered points) as ours accomplishes.
>
> - "I would recommend against saying "novel distance metric" when the proposed modification does not really satisfy triangle inequality (which metrics do)."
>
> Thank you for the comment.  We’ve shifted the phrasing to distance measure, which mimics language used in other works for distances that are not metrics (e.g. Matrix Profile series of works by Keogh et al.)

---

### Review · Reviewer_EqrV · 2025-03-13

**Summary Of Contributions:**

The manuscript introduces LINSCAN, a clustering algorithm combining DBSCAN and OPTICS, specifically designed to detect quasi-linear clusters. LINSCAN provide a novel Gaussian embedding based on local neighborhoods and a new distance function derived from KL divergence approximations. LINSCAN is empirically proven to be stable under permutations and outperforms baselines on synthetic/earthquake data, which makes it different from previous work.

**Audience:**

Yes

**Claims And Evidence:**

Yes

**Requested Changes:**

1.	A clearer, more detailed explanation of eecPts and minPts in Section 3.1.

2.	The authors should include a proof or, at the least, clarify which metric properties the distance D(P, Q) hold to help readers better understand the mathematical foundations of this distance function.

3.	Clarify the motivation and the unique advantages or differences of D(P, Q) compared to previous metrics.

4.	The experimental results analysis of LINSCAN and baselines (ADCN, DBSCAN, and OPTICS) requires further elaboration.

**Strengths And Weaknesses:**

Strengths:

1.	Instead of treating each data point as a fixed coordinate, LINSCAN embeds points as Gaussian distributions. Each point is represented by the normal distribution best approximating its local neighborhood (computed from its nearest neighbors), with the covariance matrix rescaled for consistency.

2.	The algorithm introduces a symmetric distance measure derived from an approximation of the Kullback-Leibler divergence between Gaussians. This metric is tailored to penalize differences in the covariance (and thus the local linear orientation) of the embedded points.

3.	By retaining the core principles of DBSCAN and OPTICS—such as invariance to the order of data points—LINSCAN overcomes limitations of previous methods (like ADCN) that can be sensitive to data ordering.

4.	Beyond algorithm design, the manuscript provides a rigorous theoretical analysis of the new distance function and extensive experimental results.

Weaknesses:
1.	Regarding eccPts (in LINSCAN) and minPts (in DBSCAN/OPTICS). The parameter eccPts are used without explanation. Moreover, the manuscript does not explicitly identify the commonality or differences between eccPts and minPts, and they appear to serve similar roles when considering each node’s ε-neighborhood.

2.	Regarding the distance function D(P, Q). The manuscript lacks a complete proof of whether D(P, Q) satisfies the properties of non-negativity, reflexivity, symmetry, and the triangle inequality.

3.	Regarding comparison with ADCN. Since ADCN is a density-based cluster algorithm that extends based on DBSCAN, it’s important to contain a comparison with ADCN from theoretical and methodological aspects.

4.	Visualization of comparison methods. Figure 4 presents only the results of LINSCAN under various real-world data conditions, while omitting the corresponding results from comparative methods. It may make it difficult to fully assess the advantages of LINSCAN.

---

> ### Author Response · Authors · 2025-03-21
> **Response to EqrV**
>
> We thank the reviewer for their helpful comments.  As noted in the official comment for the document, we are currently revising the manuscript to reflect these changes and will upload it within the 2 week response timeframe.  We wanted to post responses now to ensure that these will fit with your expected changes and clarify if there are any other comments / revisions / confusions in the document.
>
> - "A clearer, more detailed explanation of eecPts and minPts in Section 3.1."
>
> We’ve updated the manuscript to describe both of these parameters.  eccPts dictates the number of points to determine the local neighborhood (and thus the covariance) of a point, and minPts is the number of points needed to be connected in order to be considered a cluster instead of noise.
>
> - "The authors should include a proof or, at the least, clarify which metric properties the distance D(P, Q) hold to help readers better understand the mathematical foundations of this distance function."
>
> This was an oversight.  D(P,Q) does in fact satisfy the other requirements to be considered a metric.  We’ve updated the manuscript to add comment and proof of these other properties.
>
> - "Regarding comparison with ADCN. Since ADCN is a density-based cluster algorithm that extends based on DBSCAN, it’s important to contain a comparison with ADCN from theoretical and methodological aspects.  Clarify the motivation and the unique advantages or differences of D(P, Q) compared to previous metrics."
>
> We originally had a comparison to ADCN for a simple high-angle intersection of two clusters.  We have now added comparisons to ADCN on additional data sets, including the real world data and the synthetic random angle dataset.  We have also expanded a comment about ADCN and its downsides theoretically, including the non-symmetry of the metric and the downstream effects of this - namely that the cluster belonging is now dependent on the order of processing the points.
>
> - "The experimental results analysis of LINSCAN and baselines (ADCN, DBSCAN, and OPTICS) requires further elaboration."
>
> We have added additional comparisons to demonstrate the advantages of LINSCAN.

---

### Author Response · Authors · 2025-03-21
**Overall Review Responses**

We thank the reviewers for their helpful comments, as well as their rapid response upon submission.  There were a number of good recommendations, and we are updating the manuscript accordingly.  Overall, the major planned changes to the manuscript are:
- Addition of a larger set of experiments on real world data
- More robust comparison to existing methods to demonstrate the benefits of LINSCAN
- Expanded definitions of a few quantities, including quasi-linear clusters, our distance measure, and various hyperparameters

We are currently revising the manuscript to reflect these changes and will upload it within the 2 week response timeframe.

---

### Decision · Action_Editor_Qihn · 2025-04-13

**Recommendation:** Reject

**Comment:**

See my discussion in Claims of Evidence and in Audience above.  The contributions are for a niche application area, and unlikely to be of real interesting to the TMLR audience.  I think this would be better fit in a domain specific journal where reviewers could assess if the clusters found are actually meaningful, instead of evaluating just on the quantification that the authors devised here.

**Audience:**

While density-based cluster analysis does probably fall within TMLR audience, it is not a very active area, since most core definitions have well-explored heuristics.

This paper focuses on what reviewers see as a very niche extension of density-based clustering.  While it seems relevant for the application in seismic data, reviewers did not feel it was a compelling general extension.  A wider set of application data may have made the paper more compelling for a general audience journal (TMLR) as opposed to a domain-specific journal.

**Claims And Evidence:**

Some of the mains claims of the paper are well-supported: the near-metric analysis, the efficacy on some synthetic and specific seismic data sets.
However, other claims are not convincingly supported, namely that this method is a new technique of general interest.